# Laboratory Profile of COVID-19 Patients with Hepatitis C-Related Liver Cirrhosis

**DOI:** 10.3390/jcm11030652

**Published:** 2022-01-27

**Authors:** Bianca Cerbu, Mirela Loredana Grigoras, Felix Bratosin, Iulia Bogdan, Cosmin Citu, Adrian Vasile Bota, Madalina Timircan, Melania Lavinia Bratu, Mihaela Codrina Levai, Iosif Marincu

**Affiliations:** 1Methodological and Infectious Diseases Research Center, Department of Infectious Diseases, “Victor Babes” University of Medicine and Pharmacy, 300041 Timisoara, Romania; ionitabiancaelena@yahoo.com (B.C.); felix.bratosin7@gmail.com (F.B.); iulia.georgianabogdan@gmail.com (I.B.); citu.ioan@umft.ro (C.C.); bota.adrian1@yahoo.com (A.V.B.); timircan.madalina@yahoo.com (M.T.); bratu.lavinia@umft.ro (M.L.B.); codrinalevai@umft.ro (M.C.L.); imarincu@umft.ro (I.M.); 2Department of Psychology, “Victor Babes” University of Medicine and Pharmacy, 300041 Timisoara, Romania; 3Research Center for Medical Communication, “Victor Babes” University of Medicine and Pharmacy, 300041 Timisoara, Romania

**Keywords:** COVID-19, SARS-CoV-2 infection, hepatitis C virus, cirrhosis, ACLF, ICU admission

## Abstract

Patients with cirrhosis are known to have multiple comorbidities and impaired organ system functioning due to alterations caused by chronic liver failure. In the past two years, since the COVID-19 pandemic started, several studies have described the affinity of SARS-CoV-2 with the liver and biliary cells. Considering hepatitis C as a significant independent factor for cirrhosis in Romania, this research was built on the premises that this certain group of patients is susceptible to alterations of their serum parameters that are yet to be described, which might be useful in the management of COVID-19 in these individuals. A retrospective cohort study was developed at a tertiary hospital for infectious disease in Romania, which included a total of 242 patients with hepatitis C cirrhosis across two years, out of which 46 patients were infected with SARS-CoV-2. Stratification by patient weight and COVID-19 status identified several important laboratory serum tests as predictors for acute-on-chronic liver failure and risk for intensive care unit admission. Thus, white blood cell count, lymphocyte count, ferritin, hypoglycemia, prothrombin time, and HCV viral load were independent risk factors for ACLF in patients with COVID-19. High PT, creatinine, BUN, and HCV viral load were the strongest predictors for ICU admission. Inflammatory markers and parameters of gas exchange were also observed as risk factors for ACLF and ICU admission, including procalcitonin, CRP, IL-6, and D-dimers. Our study questions and confirms the health impact of COVID-19 on patients with cirrhosis and whether their laboratory profile significantly changes due to SARS-CoV-2 infection.

## 1. Introduction

The clinical manifestations of severe acute respiratory syndrome coronavirus type 2 (SARS-CoV-2) range from asymptomatic infection to life-threatening illness [1]. It is still uncertain why some patients infected with SARS-CoV-2 evolve to develop severe, life-threatening forms of coronavirus disease 2019 (COVID-19), whilst others have a mild or even asymptomatic clinical history. Numerous risk factors for severe COVID-19 illness have been discovered, including sociodemographic characteristics and concurrent disorders [2]. Hepatic involvement is common in SARS-CoV-2 infection, presenting clinically as increased liver function tests or sudden hepatic failure [3]. In patients with pre-existing chronic liver diseases or cirrhosis, SARS-CoV-2 infection has had inconsistent effects. Previously published small-scale studies from tertiary referral centers suggested that patients with cirrhosis who were infected with SARS-CoV-2 died at a rate of up to 40% [4]. However, other studies have shown that patients with cirrhosis who tested positive for SARS-CoV-2 infection died at a rate equivalent to those hospitalized with cirrhosis-related complications who did not test positive for SARS-CoV-2 infection [5].

In the 2018 census, more than 37 thousand new cases of hepatitis C were recorded in countries of the European Union [6], bringing the overall prevalence to 19 million cases [7] and over 70 million cases globally [8]. Since the pandemic began in 2019, there has been concern that pre-existing chronic liver disease (CLD) may predispose individuals to poor outcomes following SARS-CoV-2 infection, particularly given the overlapping risk factors for severe COVID-19 and CLD, such as advancing age, obesity, and diabetes [9,10]. The presence of a pre-existing liver condition associated with SARS-CoV-2 infection, such as hepatitis C virus (HCV), may be critical for patient prognosis, since chronic hepatitis C continues to be a health burden in a number of European nations. While the precise effect of COVID-19 on the liver is unclear, abnormalities in liver biochemistry are common in COVID-19 cases, occurring in 15–65% of SARS-CoV-2-infected individuals [11].

In our present research, we focused on determining the possible changes in serum parameters and liver function markers, which show abnormalities in routine laboratory testing, and the relationship between cirrhosis and the severity of COVID-19. We describe the possible relationship between the clinical course, laboratory findings, and the outcome of 46 patients admitted in our department with documented cirrhosis and SARS-CoV-2 infection.

## 2. Materials and Methods

We conducted a retrospective cohort study to determine the risk of severe COVID-19 infection in patients with post-hepatitis C cirrhosis who had been infected with SARS-CoV-2. We used clinical and laboratory data from patients hospitalized at Timisoara’s Infectious Diseases and Pulmonology Hospital, “Victor Babes”. The data gathering period was from 1 April 2020 to 1 December 2021. We discovered 46 instances of SARS-CoV-2 infection among 242 patients with cirrhosis who were being followed at our clinic.

Patients with hepatitis C cirrhosis exhibit unique clinical signs and symptoms such as ascites, abdominal wall vascular collaterals, hypertrophic osteoarthropathy, clubbing, and asterixis on physical examination [12]. Weakness, weariness, anorexia, and weight loss are all constitutional signs [13]. The hospitals also conducted ultrasonography and liver biopsies, and microscopically interpreted sample slides revealed evidence of regenerating nodules of hepatocytes, surrounded by fibrous connective tissue that bridges between portal tracts, confirming the diagnosis of cirrhosis. The inclusion criteria were set for patients diagnosed exclusively with non-alcoholic cirrhosis and non-overlapping diagnosis codes of chronic hepatitis B or other primary or secondary causes of cirrhosis. The diagnosis in all patients was priorly established through liver biopsy, so all patients with a secondary cause of cirrhosis other than chronic hepatitis C infection were excluded from the study after proper histopathologic immunohistochemical analysis of the liver samples [14,15].

The variables that were evaluated included patient profile data with age, sex, body mass index (BMI), medical background, cirrhosis complications (jaundice, portal hypertension, upper gastrointestinal bleeding, hepatic encephalopathy, gastritis, pleural effusion, ascites, spontaneous bacterial peritonitis, and hepatorenal syndrome), Child–Pugh staging, acute-on-chronic liver failure (ACLF), and mortality; and laboratory data with red blood cell count (RBC), platelet count (PLT), white blood cell count (WBC), neutrophils, monocytes, eosinophils, lymphocytes, hemoglobin (Hb), hematocrit (Ht), ferritin, haptoglobin, mean corpuscular volume (MCV), fasting glucose, alanine aminotransferase (ALAT), aspartate aminotransferase (ASAT), alkaline phosphatase (ALP), serum albumin, total proteins, total bilirubin, gamma glutamate transpeptidase (GGT), lactate dehydrogenase (LDH), prothrombin time (PT), partial thromboplastin time (APTT), iron, folate, vitamin B12, vitamin B6, thiamine, creatinine, blood urea nitrogen (BUN), urinary albumin, glomerular filtration rate (GFR), total cholesterol, triglycerides, VLDL-C, LDL-C, HDL-C, total lipid, procalcitonin, C-reactive protein (CRP), interleukin-6 (IL-6), tumor necrosis factor alpha (TNF-α), erythrocyte sedimentation rate (ESR), fibrinogen, arterial pH, arterial base excess, arterial oxygen pressure (pO_2_), arterial carbon dioxide pressure (pCO_2_), serum bicarbonate (HCO_3_), oxygen saturation (SaO_2_), and hepatitis C viral load.

Data of patients with post-hepatitis C cirrhosis were compared before and after SARS-CoV-2 infection using the IBM SPSS v.26 statistical software. The χ^2^ test and Fisher exact test were used for comparing proportions of categorical variables depending on the expected frequency assumption. The unpaired Student’s *t*-test was used to compare average values of normally distributed data for the COVID and non-COVID study groups. We utilized the Mann–Whitney U-test to compare median values of continuous variables and the Shapiro-Wilk test to measure the departure of data from normality. A multivariate analysis was performed to determine independent risk factors for developing ACLF and ICU admission. The significance threshold was set for α = 0.05.

The Local Commission of Ethics for Scientific Research of the “Dr. Victor Babes” Clinical Hospital for Infectious Diseases and Pulmonology in Timisoara operates under art provisions 167 of Law no. 95/2006, art. 28, chapter VIII of order 904/2006 and with EU GCP Directives 2005/28/EC, International Conference of Harmonization of Technical Requirements for Registration of Pharmaceuticals for Human Use (ICH), and with the Declaration of Helsinki—Recommendations Guiding Medical Doctors in Biomedical Research Involving Human Subjects. The current study was approved on 15 December 2021, with the approval number 12570. All study participants agreed to be involved in this study by signing an informed consent form.

## 3. Results

Patients with hepatitis C cirrhosis were divided in two groups based on their COVID-19 status at the time of study. A total of 46 patients were included in the COVID group and 196 in the non-COVID group. Table 1 describes the general characteristics of the patients involved in the study. It was noted that patients in the COVID groups had significantly lower average BMI values (*p*-value = 0.035), while the proportion of underweight patients was higher in the same COVID group (*p*-value = 0.041). More than half of our patients with cirrhosis had a history of portal hypertension, among other complications specific to their chronic liver disease, whereas hepatic encephalopathy was observed in a statistically significant higher proportion in patients infected with SARS-CoV-2 (54.3% vs. 36.2%, *p*-value = 0.023). The study participants were also grouped by the Child–Pugh score, having no significant difference in proportions of Child–Pugh A, B, or C. However, we noticed significant higher number of cirrhosis patients that developed acute-on-chronic liver failure while being infected with SARS-CoV-2 (19.6% vs. 5.1% *p*-value < 0.001). Similar findings were observed in the rate of ICU admissions in patients with COVID-19 (26.0% vs. 8.2%, *p*-value < 0.001), and in the mortality rate (15.2% vs. 6.1%, *p*-value = 0.039).

An extensive evaluation by serum laboratory analysis of our cohort is described in Table 2. The assessment of a complete blood count identified significant differences by median values and a departure from the normal range of the WBC (12,200 vs. 4600, *p*-value < 0.001), lymphocytes (6900 vs. 2500, *p*-value < 0.001), and ferritin (479 µg/L vs. 355 µg/L, *p*-value < 0.001). Liver function tests identified significant deviation from normality in fasting glucose levels (146 mmol/L vs. 128 mmol/L, *p*-value = 0.024), ALT (57 U/L vs. 44 U/L, *p*-value = 0.049), and PT (13.9 s vs. 11.2 s, *p*-value = 0.008). Major nutritional deficiency between groups was observed only in thiamine levels, where patients with COVID had significantly lower levels (2.4 µg/dL vs. 2.6 µg/dL, *p*-value = 0.041). The lipid profile did not differ between groups. Kidney function tests were significantly different in cirrhosis patients infected with SARS-CoV-2, having higher creatinine levels (1.54 µmol/L vs. 1.31 µmol/L, *p*-value < 0.001) and BUN levels (14 mmol/L vs. 11 mmol/L, *p*-value = 0.002). Finally, the HCV viral load was significantly higher in the COVID-19 group (*p*-value < 0.001).

The inflammatory profile was analyzed only in patients positive for SARS-CoV-2 infection. Therefore, data during the acute phase of COVID-19 were obtained from 46 patients, while the second evaluation was performed in 39 patients, since seven of them did not survive. Alterations of the studied parameters can be observed in Table 3. The majority of inflammatory markers were on average higher than the upper limits of the normal range both at the initial evaluation when patients were confirmed with COVID-19, as well as at 4 weeks after being tested negative for SARS-CoV-2. An important finding is the higher mean value of procalcitonin in patients at the second evaluation (0.5ug/L vs. 1.2ug/L, *p*-value < 0.001), signifying the risk of bacterial infection after the acute viral infection. The other inflammatory markers were elevated in patients during the acute phase of COVID-19, including CRP levels (56mg/L vs. 12mg/L, *p*-value < 0.001), IL-6 (49pg/mL vs. 17pg/mL, *p*-value < 0.001), fibrinogen (5.1g/L vs. 3.7g/L, *p*-value < 0.001), and D-dimers (331ng/mL vs. 262ng/mL, *p*-value < 0.001).

Arterial blood gases also showed significant alterations from the normal range. Patients with COVID-19 had statistically significant higher levels of arterial CO_2_ (49mmHg vs. 43mmHg, *p*-value < 0.001) and lower O_2_ saturation (89% vs. 92%, *p*-value = 0.028).

The clinical profile was also evaluated at 4 weeks after SARS-CoV-2 infection (Table 4), without any important findings, except pleural effusions that were found in a significantly increased number of patients (15.2% vs. 33.3% at 4 weeks, *p*-value = 0.049).

Figure 1 and Figure 2 describe the association of laboratory parameters with ACLF and ICU admission in underweight and normal weighted patients with cirrhosis from the two study groups, and the association of inflammatory markers and arterial blood gases with ACLF and ICU admission in underweight and normal weighted patients with cirrhosis from the two study groups, respectively. The odds ratio and 95% confidence intervals are developed based on BMI values that initially showed significant differences between study groups (COVID and non-COVID). The multivariate analysis described significant interactions between WBC and ferritin in all subgroups as predictors for the risk of developing ACLF. Other significant associations with ACLF were observed only in patients with COVID-19 with elevated lymphocytes and ALT levels, respectively, and low fasting glucose levels. Elevated PT, creatinine, BUN, and HCV viral load were associated with an increased risk for ICU admission in cirrhosis patients, regardless of their SARS-CoV-2 status, although the risk was higher once COVID-19 was diagnosed, and stratification by BMI indicated that underweight patients had a higher chance of being admitted to the ICU. Lastly, underweight patients with cirrhosis and COVID-19 were in the group with the highest risk factor for developing ACLF and ICU admission, where procalcitonin levels along with CRP, IL-6, and fibrinogen were all significant independent risk factors. Additionally, d-dimers, pCO_2_, and SaO_2_ were all independent risk factors for ICU admission, but not for ACLF.

## 4. Discussion

By the nature of this study, it is one of the first to describe a thorough evaluation of laboratory data in cirrhosis patients after a SARS-CoV-2 infection and to present a stratified risk analysis for developing acute-on-chronic liver failure and being admitted to an intensive care unit, while recording the nutritional status of the patient and SARS-CoV-2 infection using serum parameters. During the investigation of laboratory profiles in our cohort, we observed many instances of blood parameters’ departure from normality, where both COVID and non-COVID patients with cirrhosis showed abnormal values such as pre-existent anemia, hypoglycemia, elevated liver enzymes, poor liver function tests, nutritional deficiency, kidney failure, and hypolipidemia. Similar findings were separately described in different studies until present day, such as anemia in cirrhosis [16], hypoglycemia in cirrhosis [17], or hypolipidemia suffered by patients with cirrhosis [18]. However, all these studies did not describe their findings in conjunction with the presence of SARS-CoV-2.

As observed in the current study, patients with cirrhosis exhibit significant alterations in lipid profile compared to healthy individuals. However, SARS-CoV-2 infection in these patients did not seem to trigger important differences in serum lipid studies. Our research findings reveal that blood lipid levels, including total cholesterol, triglycerides, HDL-C, LDL-C, VLDL-C, and total lipids, are considerably lower in patients with HBV-cirrhosis, indicating hypolipidemia. Because HDL-C is generated in the liver, significant damage to hepatocytes, such as those induced by alcohol use, chronic viral hepatitis, or liver cirrhosis, may result in impaired liver function and a drop in total cholesterol and HDL-C levels, as was previously established [19]. Reduced cholesterol levels in individuals with cirrhosis showed the degree of liver cell damage, which is connected with the impairment of the liver’s capacity to synthesize [20].

Although about 30% of patients with cirrhosis have diabetes mellitus (DM) [21], it is not mediated through insulin resistance triggered by obesity. Instead, patients with cirrhosis have a tendency towards malnutrition and cachexia but can develop DM from different metabolic pathophysiologic alterations [22]. Nevertheless, the tendency towards emaciation through sarcopenia [23,24] and the cachexia of the cirrhotic patient was the reason for stratifying data in our research by the BMI of patients involved.

From the earliest reports to the present day, hemostasis abnormalities with a tendency toward hypercoagulability have been observed in COVID-19 patients [25]; in particular, increased D-dimer and fibrinogen levels have been associated with a poor prognosis in hospitalized patients, primarily due to pulmonary embolism or disseminated intravascular coagulation [26]. However, blood samples were often taken after hospitalization, and nothing is known regarding the importance of clotting parameters in identifying individuals with a more severe form of COVID-19, defined by ARDS, and their association to thromboembolic events at their initial presentation. Considering this, we evaluated all the coagulation parameters made available to our lab and found higher than normal fibrinogen, d-dimers, and PT, which were all proven individual risk factors for ICU admission, while fibrinogen alone was also a risk factor for developing ACLF. The evaluation of acute-on-chronic liver failure was another aim of this study, since it was observed that significantly more patients with cirrhosis develop this pathology [27].

Lastly, the arterial blood gases were evaluated, both due to the lung and liver affinity of the SARS-CoV-2 [28], and due to the mild hypoxemia encountered by approximately one-third of patients with chronic liver disease [29]. This anomaly can be explained by the findings that arterial venous anastomoses and communications between the portal and arterial circulations, as well as between the bronchial and pulmonary veins, are more likely to be functional in patients with cirrhosis. These anastomoses and communications account for hypoxemia and perfusion defects seen on lung scans in patients with cirrhosis. Shunts reported in individuals with severe liver illness that result in blood gas changes may be a consequence of portal hypertension development, intrapulmonary arteriovenous shunting, or ventilation–perfusion inequality [30].

## 5. Conclusions

Our study asks and confirms whether a laboratory profile of patients with cirrhosis significantly changes due to SARS-CoV-2 infection. Based on the presented results, we prove that low BMI is a good stratifier for risk analysis in developing acute-on-chronic liver failure and being admitted to the intensive care unit. Several inflammatory markers, such as procalcitonin, CRP, IL-6, fibrinogen, and d-dimers, show strong interactions with SARS-CoV-2 infection, being significant risk factors for ICU admission. Nevertheless, the laboratory profile of patients with cirrhosis is altered by the intrinsic mechanisms of cirrhosis, and it is still being modified further by the presence of the SARS-CoV-2 virus, indicating that WBC, lymphocyte count, creatinine, ferritin, BUN, and ALT levels rise significantly in patients with COVID-19.

## Figures and Tables

**Figure 1 jcm-11-00652-f001:**
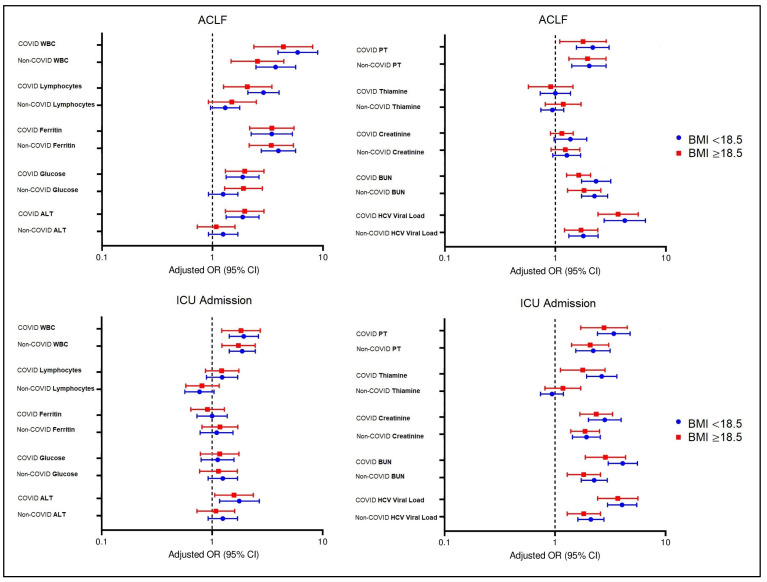
The association of laboratory parameters with ACLF and ICU admission in underweight and normal weighted patients with cirrhosis from the two study groups.

**Figure 2 jcm-11-00652-f002:**
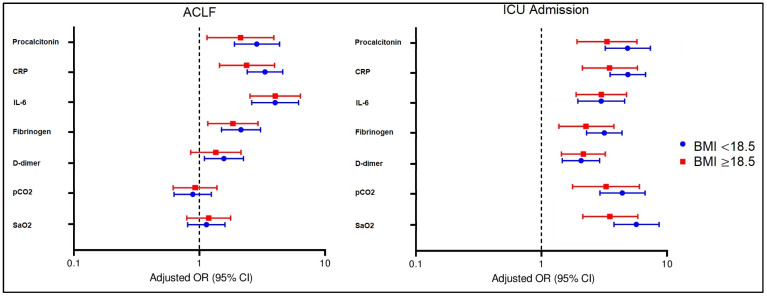
The association of inflammatory markers and arterial blood gases with ACLF and ICU admission in underweight and normal weighted patients with cirrhosis from the two study groups.

**Table 1 jcm-11-00652-t001:** Characteristics of study cohort compared with non-COVID patients with cirrhosis.

Characteristics *	COVID (*n* = 46)	Non-COVID (*n* = 196)	*p*-Value **
Age, years			0.291
18–65	34 (73.9%)	129 (65.8%)	
>65	12 (26.1%)	67 (34.2%)	
**Sex**			0.751
Men	27 (58.7%)	110 (56.1%)	
Women	19 (41.3%)	86 (43.9%)	
BMI (mean ± SD)	20.2 ± 2.1	21.4 ± 3.7	0.035^¶^
Underweight (<18.5 kg/m^2^)	15 (32.6%)	37 (18.8%)	0.041
Medical Background			
Chronic alcohol consumption	16 (34.8%)	61 (31.1%)	0.631
History of hepatitis A infection	7 (15.2%)	19 (9.7%)	0.276
Cardiovascular disease	28 (60.9%)	125 (63.8%)	0.712
Metabolic disease	11 (23.9%)	51 (26.0%)	0.768
Cerebrovascular disease	11 (23.9%)	47 (23.9%)	0.992
Chronic kidney disease	9 (19.6%)	40 (20.4%)	0.898
Malignancy	6 (13.0%)	31 (15.8%)	0.638
Cirrhosis complications			
Jaundice	13 (28.2%)	48 (24.5%)	0.596
Portal hypertension	28 (60.9%)	103 (52.6%)	0.308
Upper gastrointestinal bleeding	19 (41.3%)	68 (34.7%)	0.400
Hepatic encephalopathy	25 (54.3%)	71 (36.2%)	0.023
Gastritis	27 (58.7%)	99 (50.5%)	0.317
Pleural effusion	7 (15.2%)	33 (16.8%)	0.790
Ascites	19 (41.3%)	67 (34.2%)	0.363
Spontaneous bacterial peritonitis	4 (8.7%)	8 (4.1%)	0.194
Hepatorenal syndrome	4 (8.7%)	7 (3.6%)	0.133
Child–Pugh			0.542
A	26 (56.5%)	115 (58.7%)	
B	13 (28.2%)	42 (21.4%)	
C	7 (15.2%)	39 (19.9%)	
Disease outcomes			
ACLF	9 (19.6%)	10 (5.1%)	<0.001
ICU admission	12 (26.0%)	16 (8.2%)	<0.001
Mortality	7 (15.2%)	12 (6.1%)	0.039

* Data reported as *n* (%), unless specified differently; ** Chi-square test and Fisher’s exact; ^¶^ Unpaired Student’s *t*-test; ACLF—Acute-on-chronic liver failure≥.

**Table 2 jcm-11-00652-t002:** Comparison of serum parameters between COVID-19 and non-COVID-19 patients with cirrhosis.

Variables *	Normal Range	COVID (*n* = 46)	% Outside Normality	Non-COVID (*n* = 196)	% Outside Normality	*p*-Value **
Complete blood count						
RBC (millions/mm^3^)	4.35–5.65	3.31 ^ (1.5)	70.4%	3.39 ^ (1.3)	67.2%	0.301
PLT (thousands/mm^3^)	150–450	94 ^ (116)	75.1%	90 ^ (98)	71.0%	0.442
WBC (thousands/mm^3^)	4.5–11.0	12.2 ^ (7.3)	48.1%	4.6 (2.6)	22.4%	<0.001
Neutrophils (thousands/mm^3^)	1.5–8.0	5.0 (5.3)	16.1%	3.8 (4.0)	20.7%	0.056
Monocytes (thousands/mm^3^)	0.1–1.0	0.4 (0.5)	7.8%	0.6 (0.3)	9.2%	0.840
Eosinophils (units/mm^3^)	30–300	142 (96)	2.2%	173 (101)	4.9%	0.727
Lymphocytes (thousands/mm^3^)	1.0–4.8	6.9 ^ (6.2)	63.6%	2.5 (3.6)	38.8%	<0.001
Hb (g/dL)	13.0–17.0	11.8 ^ (4.6)	67.2%	12.0 ^ (5.2)	65.6%	0.882
Hematocrit (%)	36–48	37 (12)	22.4%	39 (13)	20.5%	0.904
Ferritin (µg/L)	24–336	479 ^ (301)	88.1%	355 ^ (205)	52.2%	<0.001
Haptoglobin (mg/dL)	41–165	72 (52)	35.3%	79 (64)	31.6%	0.651
Mean corpuscular volume (fL)	80–96	88 (94)	32.7%	85 (91)	34.5%	0.629
Liver function tests						
Fasting glucose (mmol/L)	60–125	146 ^ (84)	73.5%	128 ^ (80)	62.4%	0.024
ALT (U/L)	7–35	57 ^ (42)	68.2%	44 ^ (35)	54.7%	0.049
AST (U/L)	10–40	43 ^ (36)	35.5%	36 (30)	28.0%	0.063
ALP (U/L)	40–130	141 ^ (110)	53.6%	138 ^ (106)	48.4%	0.474
Serum albumin (g/dL)	3.4–5.4	3.6 (1.2)	15.3%	3.8 (1.4)	12.9%	0.703
Total proteins (g/dL)	6.0–8.3	6.2 (3.5)	17.8%	6.1 (2.3)	17.1%	0.936
Total bilirubin (g/dL)	0.3–1.2	1.5 ^ (1.6)	52.0%	1.4 ^ (1.2)	48.3%	0.522
GGT (U/L)	0–30	48 ^ (34)	73.1%	46 ^ (30)	66.5%	0.162
LDH (U/L)	140–280	245 (144)	23.5%	267 (151)	26.8%	0.329
PT (seconds)	11.0–13.5	13.9 ^ (7.5)	40.3%	11.2 (4.2)	24.6%	0.008
APTT (seconds)	30–40	39 (12)	23.6%	36 (8)	13.0%	0.101
Nutritional deficiency						
Iron (µg/dL)	60–170	64 (32)	37.4%	62 (32)	38.9%	0.946
Folate (nmol/mL)	2.7–17.0	2.9 (2.0)	31.0%	3.0 (2.1)	32.4%	0.951
Vitamin B12 (pg/mL)	160–950	167 (237)	44.6%	171 (222)	40.3%	0.704
Vitamin B6 (µg/L)	5–50	8 (11)	35.5%	8 (14)	36.9%	0.892
Thiamine (µg/dL)	2.5–7.5	2.4 ^ (3.1)	59.6%	2.6 (2.3)	48.3%	0.041
Kidney function tests						
Creatinine (µmol/L)	0.74–1.35	1.54 ^ (2.62)	68.7%	1.31 (1.50)	50.7%	<0.001
BUN (mmol/L)	2.1–8.5	14 ^ (16)	75.5%	11 ^ (11)	62.2%	0.002
Urinary albumin (mg/g)	0–30	42 ^ (13)	62.6%	40 ^ (10)	59.3%	0.776
GFR	>60	49 ^ (32)	73.0%	56 ^ (26)	64.9%	0.055
Lipid profile						
Total cholesterol (mg/dL)	100–200	101.5 (60.3)	52.7%	112 (54.6)	48.6%	0.266
Triglycerides	50–150	66.4 (41.7)	17.3%	69.2 (39.4)	15.1%	0.826
VLDL-C (mg/dL)	2–30	17.1 (9.5)	7.2%	18.2 (9.0)	7.6%	0.943
LDL-C (mg/dL)	<100	79.2 (36.8)	11.6%	76.1 (38.4)	9.9%	0.794
HDL-C (mg/dL)	40–60	34.0 ^ (19.3)	35.4%	39.5 ^ (21.4)	27.4%	0.057
HCV viral load (U/L × 10^3^)	<15	38,402 (35,195)	100%	29,365 (24,392)	100%	<0.001

* Data reported as median (IQR), unless specified differently; ** Mann–Whitney U-test; ^ median value outside the normal range.

**Table 3 jcm-11-00652-t003:** Dynamics in inflammatory markers and blood gases of patients with cirrhosis at 4 weeks after SARS-CoV-2 infection.

Variables *	Normal Range	During COVID (*n* = 46)	After COVID (*n* = 39)	*p*-Value ***
Inflammatory markers				
Procalcitonin (ug/L) **	0–0.5 ug/L	0.5 ± 0.2	^ 1.2 ± 1.0	<0.001
CRP (mg/L) **	0–10 mg/L	^ 56 ± 13	^ 12 ± 7	<0.001
IL-6 (pg/mL) **	0–16 pg/mL	^ 49 ± 17	^ 17 ± 10	<0.001
TNF-α (pg/mL)	0–29 pg/mL	^ 42 ± 9	^ 39 ± 8	0.111
IFN-γ (pg/mL)	0–3 pg/mL	^ 3.2 ± 0.5	3.0 ± 0.6	0.097
ESR (mm/h)	0–22 mm/hr	^ 43 ± 8	^ 40 ± 8	0.088
Fibrinogen (g/L)	2–4 g/L	^ 5.1 ± 1.0	3.7 ± 1.0	<0.001
D-dimer (ng/mL)	<250	^ 331 ± 53	^ 262 ± 31	<0.001
Arterial blood gas				
Arterial pH	7.35–7.45	^ 7.46 ± 0.6	7.37 ± 0.9	0.584
pO_2_ (mmHg)	80–100 mmHg	^ 77 ± 12	80 ± 9	0.202
pCO_2_ (mmHg)	35–45 mmHg	^ 49 ± 5	43 ± 7	<0.001
HCO_3_ (mEq/L)	22–28 mEq/L	23 ± 6	24 ± 4	0.377
SaO_2 (%)_	94–100%	^ 89 ± 7	^ 92 ± 5	0.028

* Data reported as mean ± SD, unless specified differently; ** Mann–Whitney U-test; *** paired *t*-test; ^ data outside the normal range.

**Table 4 jcm-11-00652-t004:** Reevaluation of patients with cirrhosis at 4 weeks after SARS-CoV-2 infection.

Clinical Outcomes *	Acute COVID (*n* = 46)	After COVID (*n* = 39)	*p*-Value **
Jaundice	13 (28.3%)	16 (41.0%)	0.216
Portal hypertension	28 (60.9%)	28 (71.8%)	0.289
Upper gastrointestinal bleeding	6 (13.0%)	2 (5.1%)	0.497
Encephalopathy	25 (54.3%)	24 (61.5%)	0.503
Gastritis	27 (58.7%)	24 (61.5%)	0.789
Pleural effusion	7 (15.2%)	13 (33.3%)	0.049
Ascites	19 (41.3%)	21 (53.9%)	0.248
Spontaneous bacterial peritonitis	4 (8.7%)	7 (17.9%)	0.205
Hepatorenal syndrome	4 (8.7%)	3 (7.7%)	0.866
Child–Pugh Score			0.543
Child–Pugh A	26 (56.5%)	25 (64.1%)	
Child–Pugh B	13 (28.3%)	11 (28.2%)	
Child–Pugh C	7 (15.2%)	3 (7.7%)	

* Data reported as n (frequency); ** Chi-square test and Fisher’s exact.

## Data Availability

Data available on request.

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
