# Peer review of "Laboratory Profile of COVID-19 Patients with Hepatitis C-Related Liver Cirrhosis"

_jcm, 2022, doi:10.3390/jcm11030652_

Round 1

Reviewer 1 Report

Well-designed study which presents a novel aspect of COVID-19 effect on cirrhotic patients due to HCV infection. 

Well-written, comprehensive  and concise.

Kindly re-consider the value of including ref 15. I would suggest to omit it.  

Author Response

Dear reviewer,

Thank you for your precious feedback. We appreciate the time and effort invested in carefully analysing our manuscript.

There were some slight mistakes in our reported data that had to be changed as follows:

  1. Table 1: we corrected the age distribution, where non-COVID patients aged >65 were 67 (34.2%), and the corresponding percentage = 0.291.
  2. Table 1: the proportion of underweight patients was reported erroneously. There were 37 (18.8%) underweight patients in the non-COVID group. P-value=0.041
  3. Table 1: we recalculated the p-value for mortality, and changed it to 0.039 instead of the previous 0.041
  4. Table 1: we rounded off correctly all percentages
  5. Table 3: we identified procalcitonin, CRP, and IL-6 to have non-normal distributions. Since sample size was also small, we decided to test again using the Mann-Whitney test, but the p-value remains highly statistically significant.
  6. Materials and Methods: we fixed the statistical analysis description (Lines 107-112).
  7. We replaced the reference number 15, as suggested.

Best regards,

The authors

Reviewer 2 Report

Dear Author,

I appreciate your work, however, it needs improvement specially in results and analysis. Please find my comments below; 

  1. In table 1, non-covid cases were 196. However under age distribution it was presented as 129 (18-65 yr) and 71 (>65 yr), which is 200 (129+71). Corresponding percentage was also incorrect. Please clarify.
  2. In table 1, Underweight proportion was 15 (32.6%) and 38 (19.3%) for covid and non-covid. The mentioned percentage was not correct also the p value. The p value is 0.051 which is not significant. Please clarify.
  3. In case of mortality again the p value is not correct; it is 0.039 rather than 0.041.
  4. I suggest re calculating all the percentage and rounding off correctly.  
  5. In table 2, Mann-whitney U test was used for all the variables. Are all parameters not normally distributed? Its better if you can provide mean (sd) values also, which may help future researcher in terms of sample size estimation etc.
  6. In table 3, paired t test was used for all the parameters, however in some variables like (procalcitonin, CRP, IL6) standard deviation is high which may indicate non normal distribution of the data. Please check and confirm.
  7. Multivariable analysis was used but it does not clear that how variables were selected for the multivariable analysis?
  8. The statistical analysis section in the methodology section (page 104-111) is not complete. Please mention all the used analysis and describe in detail.

Author Response

Dear reviewer,

Thank you for your precious feedback. We appreciate the time and effort invested in carefully analysing our manuscript.

Please find below the edits made based on your recommendations:

  1. Table 1: we corrected the age distribution, where non-COVID patients aged >65 were 67 (34.2%), and the corresponding percentage = 0.291.
  2. Table 1: the proportion of underweight patients was reported erroneously. There were 37 (18.8%) underweight patients in the non-COVID group. P-value=0.041
  3. Table 1: we recalculated the p-value for mortality, and changed it to 0.039 instead of the previous 0.041
  4. Table 1: we rounded off correctly all percentages
  5. Table 2: we considered the Mann-Whitney U-test a better estimate of serum parameters in conjunction with the columns describing the percentage outside normality.
  6. Table 3: we identified procalcitonin, CRP, and IL-6 to have non-normal distributions. Since sample size was also small, we decided to test again using the Mann-Whitney test, but the p-value remains highly statistically significant.
  7. All variables were included simultaneously into the equation for multiple regression. The significant predictors based on p-value probability were then included in the graphical representation, even though not all confidence intervals were significant by range.
  8. Materials and Methods: we fixed the statistical analysis description (Lines 107-112).

Best regards,

The authors